# Production of constrained L-cyclo-tetrapeptides by epimerization-resistant direct aminolysis

Huan Chen [1,3], Yuchen Zhang[2,3], Yuming Wen [1], Xinhao Fan[1], Nicholas Sciolino [1], Yanyun Lin[1], Leonard Breindel[1], Yuanwei Dai[1], Alexander Shekhtman [1] ✉, Xiao-Song Xue [2] ✉ & Qiang Zhang [1] ✉

The synthesis of constrained 12-membered rings is notably difficult. The main challenges result from constraints during the linear peptide cyclization. Attempts to overcome constraints through excessive activation frequently cause peptidyl epimerization, while insufficient activation of the C-terminus hampers cyclization and promotes intermolecular oligomer formation. We present a β-thiolactone framework that enables the synthesis of cyclo-tetrapeptides via direct aminolysis. This tactic utilizes a mechanism that restricts C-terminal carbonyl rotation while maintaining high reactivity, thereby enabling efficient head-to-tail amidation, reducing oligomerization, and preventing epimerization. A broad range of challenging cyclo-tetrapeptides ( > 20 examples) are synthesized in buffer and exhibits excellent tolerance toward nearly all proteinogenic amino acids. Previously unattainable macrocycles, such as cyclo-L-(Pro-Tyr-Pro-Val), have been produced and identified as μ-opioid receptor (MOR) agonists, with an $EC_{50}$ value of 2.5 nM. Non-epimerizable direct aminolysis offers a practical solution for constrained peptide cyclization, and the discovery of MOR agonist activity highlights the importance of overcoming synthetic challenges for therapeutic development.

Head-to-tail peptide amidation is a widely adopted strategy for the synthesis of cyclopeptides[1]. Orthogonally protected linear precursors undergo cyclization through activation of the C-terminal carbonyl via potent reagents in organic solvents[2]. Alternatively, in an aqueous/organic mixed solution, direct aminolysis between the N-terminal amino group and C-terminal thioester (or its equivalent) can achieve efficient head-to-tail cyclization, leading to the formation of cyclopeptides[3–5]. The efficiency of head-to-tail cyclization for smaller macrocycles diminishes significantly because of ring strain. Adverse events are pronounced during the construction of proteinogenic L-cyclic tetrapeptides containing a constrained 12-membered ring. Insufficient peptide C-terminus activation hinders cyclization and

promotes intermolecular oligomerization[6], but overcoming ring closure constraints through excessive activation leads to undesired epimerization (Fig. 1a). Various strategies, including ring contraction[7–13], cyclization inducers[14–17], metal templates[18–21], and the use of D-amino acids[22], noncanonical amino acids, and artificial functional groups[23], have been employed for the synthesis of specific tetracyclic peptides and cyclopeptide analogs. However, these approaches are insufficient for developing a practical approach to accessing diverse proteinogenic cyclic tetrapeptide motifs due to limitations in amino acid scope and reliance on noncanonical or selected amino acids[1,2]. Our NPC strategy fails to cyclize linear tetrapeptide at all[24]. Despite their pharmacological promise, the study of cyclic tetrapeptides and the scope of

[1]Department of Chemistry, State University of New York, University at Albany, Albany, NY 12222, USA. [2]Key Laboratory of Fluorine and Nitrogen Chemistry and Advanced Materials, Shanghai Institute of Organic Chemistry, University of Chinese Academy of Sciences, Chinese Academy of Sciences, 345 Lingling Road, 200032 Shanghai, China. [3]These authors contributed equally: Huan Chen, Yuchen Zhang. ✉e-mail: ashekhtman@albany.edu; xuexs@sioc.ac.cn; qzhang5@albany.edu

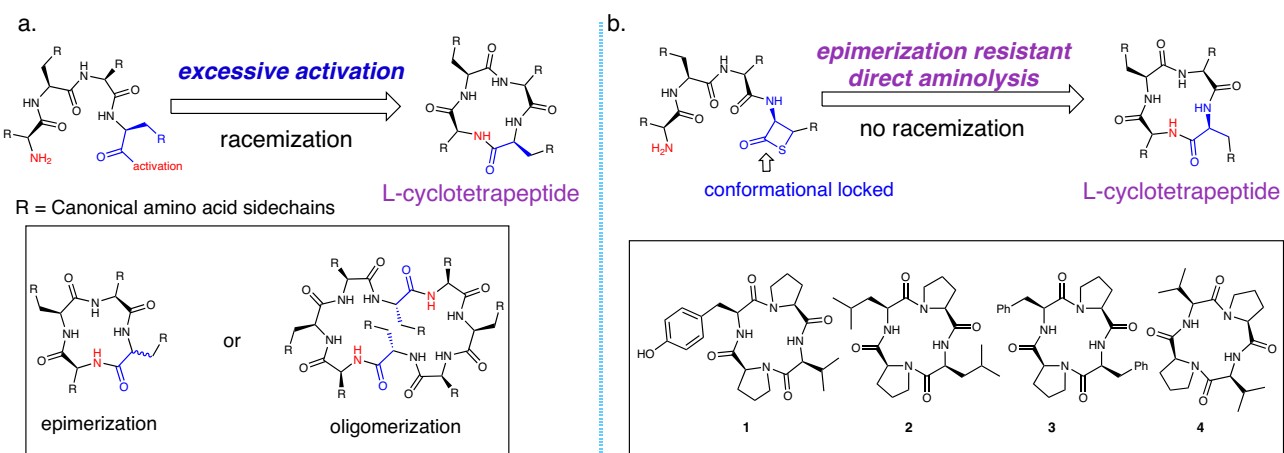

**Fig. 1 | Epimerization-resistance direct aminolysis mediated proteinogenic tetrapeptide cyclization. a** Extant activation mode. **b** β-Thiolactone facilitated tetracyclic peptide formation (This study). Inset: Challenging-to-synthesize L-cyclic tetrapeptides.

related research are hampered by limited synthetic capabilities[25,26]. Nonetheless, cyclic tetrapeptides, regarded as "privilege molecules,"[27,28] possess constrained motifs, adhere to Lipinski[29] rules, and exhibit promise for improved interactions with biomolecules[30–32] and blood–brain barrier penetration[33].

This study presents an approach designed to achieve cyclic tetrapeptide preparation utilizing β-thiolactone scaffold as a C-terminus promoter. The release of the thietane ring strain enables efficient head-to-tail aminolysis and facilitates strained ring closure in an aqueous buffer. Notably, the thietane ring's conformational restriction completely inhibited epimerization and minimized intermolecular dimerization via the epimerization-resistant direct aminolysis process. This approach yields diverse all-L-cyclo-tetrapeptides with high reactivity and minimal epimerization and oligomerization. The β-thiolactone strategy represents a significant advancement in mitigating epimerization and serves as a versatile approach for the synthesis of cyclo-tetrapeptides.

## Results
### Cyclization of cyclic tetrapeptides
The initial application of the β-thiolactone strategy involved the synthesis of challenging proteinogenic cyclic tetrapeptides with the consensus sequence L-cyclo(Pro-Xxx-Pro-Xxx)[28], in which Xxx represents Val, Leu, or Phe (**1**–**4**; Fig. 1b). Specifically, the syntheses of L-cyclo(Pro-Leu-Pro-Leu) and L-cyclo(Pro-Val-Pro-Val) have been reported with abysmal yields of 5%[34] and 7%[35], respectively. Furthermore, the tyrosinase inhibitor L-cyclo(Pro-Tyr-Pro-Val) has not been successfully synthesized since 1993, leading to its widely regarded status as unattainable[25,36]. Synthetic attempts using conventional coupling strategies did not yield L-cyclo(Pro-Tyr-Pro-Val). Instead, cyclic tetrapeptide with D-amino acid residues[37] or its triazole analog[38] was obtained. Furthermore, acyl migration as an alternative tactic failed to produce cyclic tetrapeptide due to structural constraints[39].

The head-to-tail linkage in cyclic peptides **1**–**4** (cf. Fig. 1) involves coupling a proline residue with sterically hindered amino acids (Val, Leu, Phe), which presents synthetic obstacles due to ring strain and steric hindrance at the coupling sites. We believe that incorporating β-thiolactones at the peptide C-terminus can address these issues[40]. The intramolecular cyclization of linear tetrapeptide **5**, involving direct amidation between the N-terminal proline and C-terminal β-thiolactone, yielded cyclo-tetrapeptide analog **6** (Fig. 2). The cyclization of peptide **5** was further investigated using different conditions. At a concentration of 1 mM and ambient temperature, pure water in the Falcon conical tube did not facilitate the desired reaction. However, when peptide **5** was placed in a thin borosilicate glass vial with pure

water, the desired cyclo-oligomer **6** was obtained (49%) after 30 h. Considering the borosilicate glass vial might contain cations, such as Li[41], Na[42], Ag[43], Ni, Pd, and Cu[19], a comprehensive investigation of various salts[44], including LiCl, KCl, MgCl$_2$, CaCl$_2$, Al$_2$O$_3$, SiO$_2$, B$_2$O$_3$, NaCl, NaH$_2$PO$_4$, Na$_2$HPO$_4$, K$_2$HPO$_4$, and NaOH (see Supplementary Information for detail)[45], was performed to explore their roles in cyclization process. Interestingly, employing a stoichiometric amount of Na$_2$B$_4$O$_7$ (Borax) as an additive led to optimized results, with the formation of the desired cyclo-monomer **6** after 8.5 h. Macrocycle **6** was obtained in good yield (65%) with minimal oligomerization (**6**/**6b** = 1/0.03). The exact role of borax in the cyclization mechanism is unclear, but borax or its main gradient may act as a bifunctional catalyst, facilitating the activation of β-thiolactone and the N-terminal amine[45]. To further under the role of borax reagent, preliminary [11]B NMR studies were conducted (Fig. 2a). Borax (Na$_2$[B$_4$O$_5$(OH)$_4$]·8H$_2$O) supplies 4 equivalents of boron per molar molecule. Borax (1 equivalent, 1 mM) in D$_2$O showed two peaks at 2 and 20 ppm, representing the boron complex and B(OH)$_4$⁻, respectively. Addition of an equal molar ratio of peptide **5** did not induce any spectral shifts or new peaks during the 5-min to 8-h observation period. Superimposing the spectra before and after peptide addition revealed no discernible changes, indicating no interaction between the peptide and borax. The influence of borax concentration on cyclization was investigated. At different concentrations (0–1 equivalent of borax), product and byproduct yields showed no significant differences. Moreover, the kinetic plot of the reaction, utilizing the equation[46] [−ln (1−%yield)] against time at different equivalents of borax. the kinetic plot using varied concentrations revealed distinct rate trends (Fig. 2b). Slope calculations suggest that at 1 molar equivalent of borax, cyclization was markedly accelerated, whereas 0.5 equivalent borax displayed a flatter slope. Conversely, there was no substantial rate disparity observed among 0.25, 0.125, and 0 equivalents of borax. Overall, stoichiometric borax (1 equivalent) accelerated cyclization as a first-order reaction, albeit its catalytic effect was weak (see Supplementary Information Section VI for details); however, without the addition of borax, lower yields (by 10%) and greater amounts of oligomer **6b** were observed at identical pH values (entry 7). In contrast, cyclization reactions carried out under different pH conditions, either lower or higher, resulted in decreased yields and increased production of dimeric byproducts (entries 8 and 9).

In general, the reaction yields are influenced by both pH and the addition of borax. The presence of TFA additive in preparative HPLC eluents leads to the isolation of the linear peptide in its TFA salt form. The acidic nature of the linear peptide affects the reaction pH. The pH measured after adding the acidic peptide **5** to the Falcon tube is

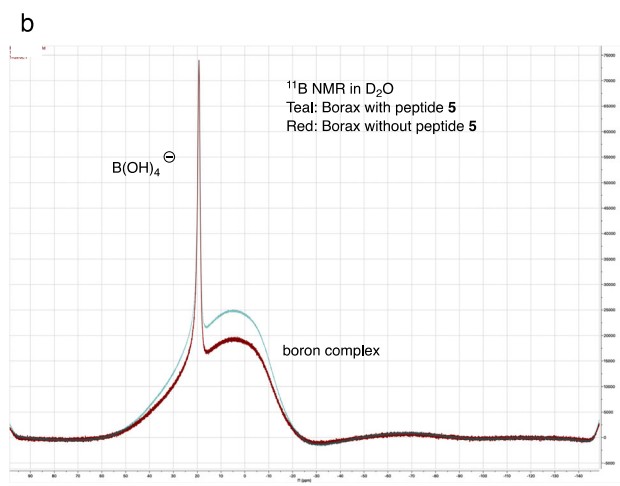

| Solution pH | Solution pH with 5 | Reaction Time | Yield of 6[c] | Yield of 6b |
|---|---|---|---|---|
| 6.00 (pure H₂O in Falcon tube) | 2.90 | 24.0 h | 1% | N.D. |
| 7.06 (H₂O in borosilicate glass) | 6.98 | 30.0 h | 49% | 1.5% |
| 7.45 (PBS/1.0 equiv. Borax) | 7.42 | 8.5 h | 66.2% | 2.1% |
| 7.45 (PBS/0.5 equiv. Borax) | 7.42 | 7.8 h | 66.0% | 1.9% |
| 7.45 (PBS/0.25 equiv. Borax) | 7.43 | 8.1 h | 57.0% | 2.75% |
| 7.45 (PBS/0.125 equiv. Borax) | 7.44 | 8.1 h | 60.5% | 2.71% |
| 7.45 (PBS without Borax) | 7.44 | 8.2 h | 56.3% | 4.0% |
| 7.14 (PBS/1.0 equiv. Borax) | 7.12 | 21.3 h | 62.0% | 7.7% |
| 8.04 (PBS/1.0 equiv. Borax) | 7.98 | 4.0 h | 58.3% | 9.3% |

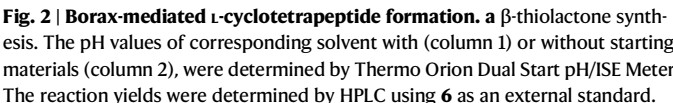

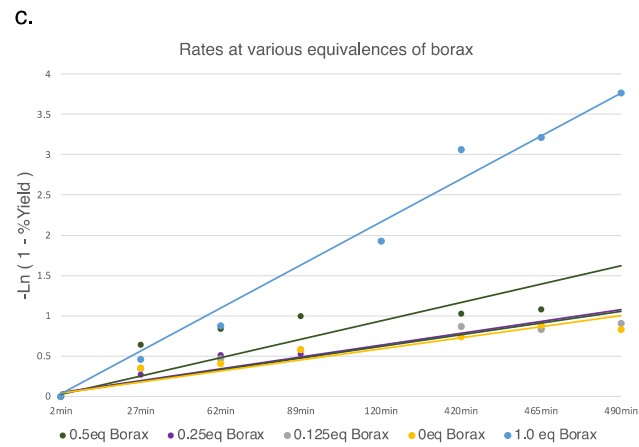

**Fig. 2 | Borax-mediated L-cyclotetrapeptide formation. a** β-thiolactone synthesis. The pH values of corresponding solvent with (column 1) or without starting materials (column 2), were determined by Thermo Orion Dual Start pH/ISE Meter. The reaction yields were determined by HPLC using **6** as an external standard.

Borosilicate glass was referred to Fisherbrand™ scintillation vials. **b** Overlaying the borax spectra before and after peptide addition in D₂O. **c** The kinetic plot of the reaction, utilizing the equation [−ln (1−%yield)] against time at different equivalents of borax.

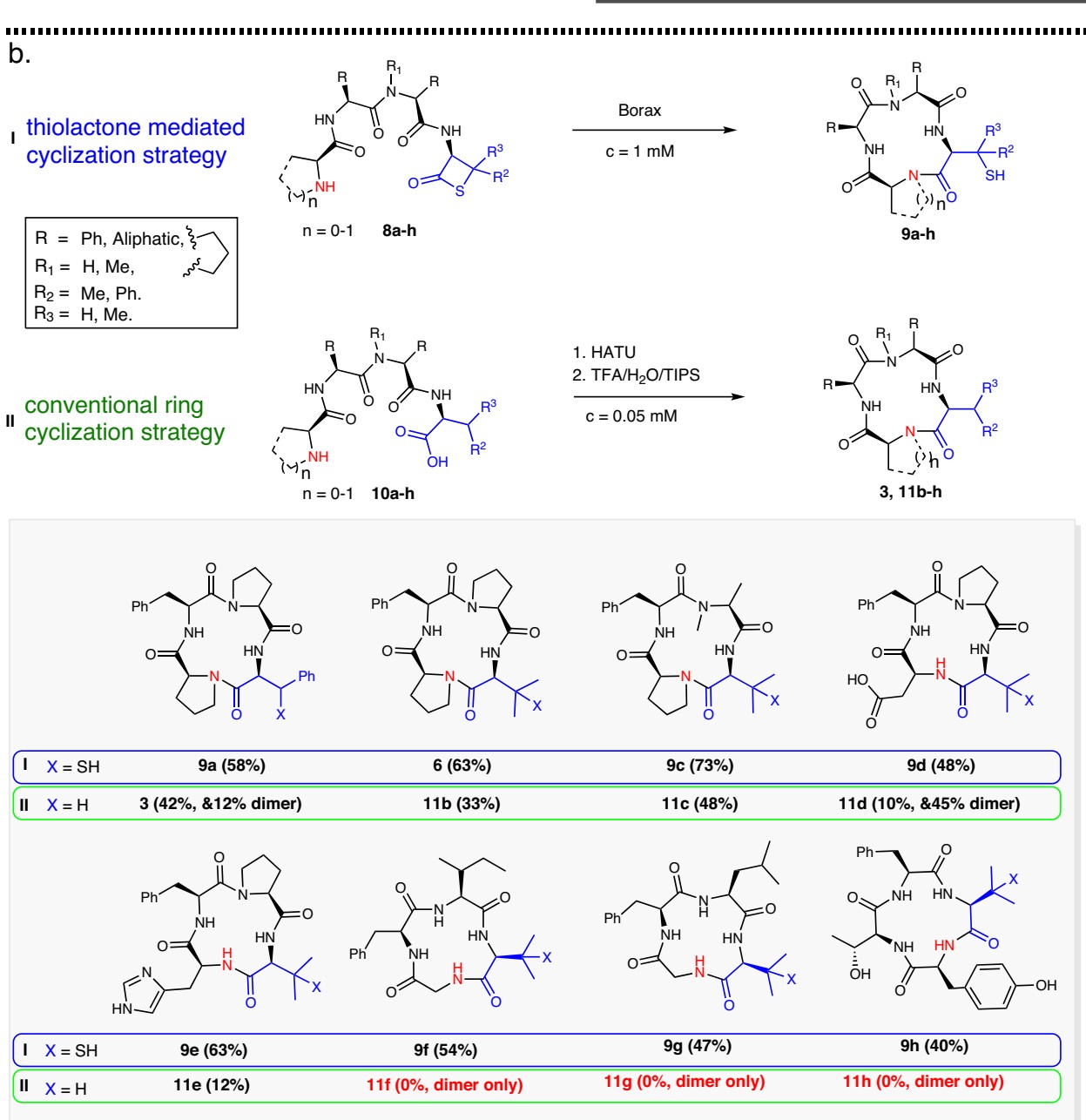

**Fig. 3 | Cyclization strategies evaluation. a** β-Thiolactone synthesis. Reaction conditions: see SI Section II. **b** Cyclization strategy comparison. All yields are HPLC isolation yields. thiolactone mediated cyclization conditions: PBS Buffer, pH = 7.4, borax (1.0 equiv), rt, *c* = 1 mM. Conventional ring cyclization conditions: (i) HATU, DIPEA, rt, DMF/dichloromethane, *c* = 0.05 mM; (ii) TFA/H₂O/TIPS = 95/2.5/2.5 (V/V/V).

around 3, while it remains close to neutral in the thin borosilicate glass vial. A slightly basic conditions enables the cyclization. Meanwhile, borax is critical for enhancing reaction yields, without the addition of borax, cyclization suffers reducing yield and substantial oligomer formation. The optimized conditions (1.0 equivalent of borax in PBS buffer, pH = 7.45) efficiently furnish cyclic tetrapeptide via direct

aminolysis between proline and β-thiolactone, resulting in minimal polymer formation and the elimination of hydrolyzed products.

The synthesis of β-thiolactones is straightforward, and the TFA salts of β-thiolactones **7a–d** can be prepared in two steps, as described in Fig. 3a[47]. With the established coupling method[38], linear precursors **8a–h** containing Val- or Phe-β-thiolactone at the C-terminus were

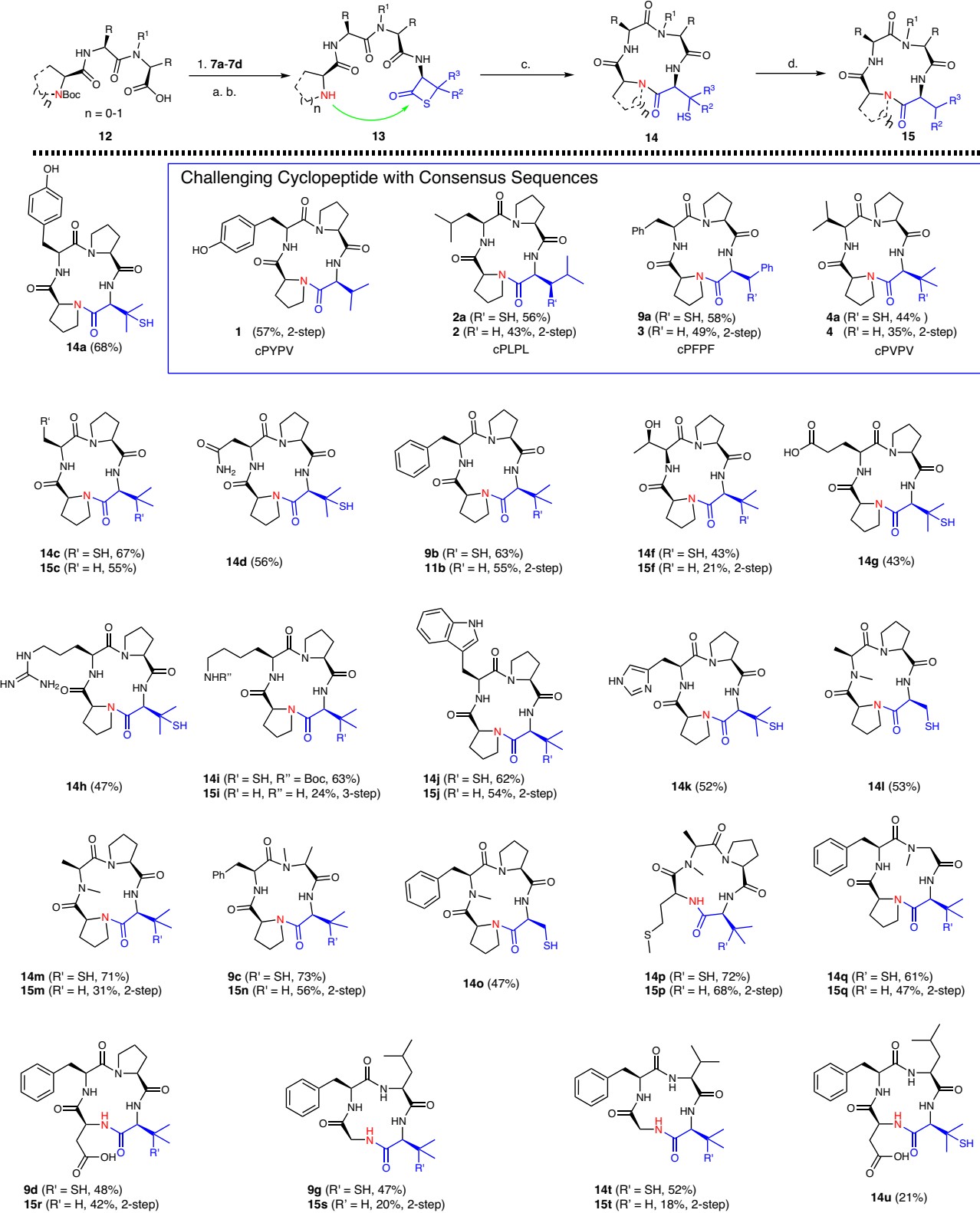

**Fig. 4 | Cyclization scope of all L-cyclopeptides.** All yields are HPLC isolation yields. R = Ph, Aliphatic, $R_1$ = H, Me, $R_2$ = Me, Ph, $R_3$ = H, Me. **a** EDC (2.5 equiv.), HOOBt (2.5 equiv.), $CH_2Cl_2$, −10 °C, 1 h. **b** TFA/TIPSH/$H_2O$ = 95/2.5/2.5 (V/V/V), rt, 20 min. **c** Borax (1.0 equiv.), PBS Buffer, rt, $c$ = 1 mM, pH = 7.4, 4–8 h. **d** VA-044 (6.0 mM), TCEP (0.07 M), glutathione (0.07 M), 37 °C, 1 h, 2-step one-pot. note: cPYPV (cyclo-Proline-Tyrosine-Proline-Valine), cPLPL (cyclo-Proline-Leucine-Proline-Leucine), cPFPF (cyclo-Proline-Phenylalanine-Proline-Phenylalanine), cPVPV (cyclo-Valine-Proline-Valine).

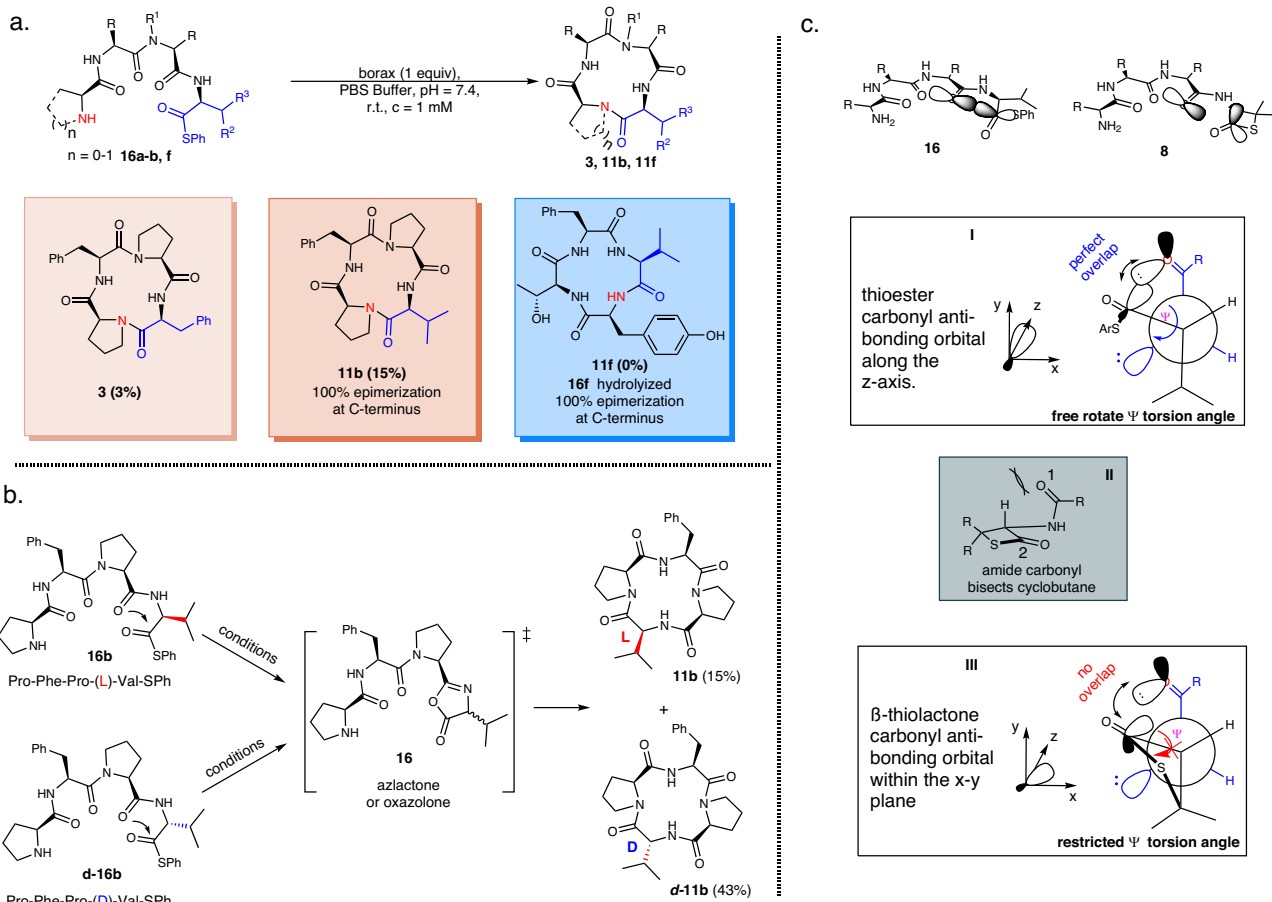

**Fig. 5 | Cyclization strategies evaluation. a** Epimerization of direct aminolysis. **b** Mechanism of epimerization. Conditions: borax (1.0 equiv.), PBS Buffer, rt, $c = 1$ mM, pH = 7.4, 4–8 h. **c** Conformation lock prevents epimerization. R = Ph, Aliphatic, $R_1$ = H, Me, $R_2$ = Me, Ph, $R_3$ = H, Me.

obtained. Under optimized cyclization conditions, ring closure of these precursors (**8a–h**) resulted in the formation of desired cyclic tetrapeptides **9a–h** without any complications (Fig. 3b). To assess the efficacy of the β-thiolactone approach against conventional methods, analogous constrained sequences were cyclized in parallel using HATU and DIPEA in dimethylformamide/dichloromethane[48]. Orthogonally protected linear peptides **10a–h** were subjected to HATU coupling conditions, affording products **3** and **11b–h** in various outcomes (Fig. 3c). Macrocycles containing two prolines were obtained in good yields (**3**, 42%; **11b**, 33%), although **3** exhibited 12% dimer formation. In contrast, the β-thiolactone approach yielded similar constructs with higher yields and no oligomerization (**9a**, 58%; **6**, 63%). Moreover, the HATU coupling method performed poorly for cyclopeptides that contained a single proline or N-methyl moiety (**11c**, **11d**, **11e**), resulting in significantly lower yields (48%, 10%, and 12%) and extensive polymerization of **11d** (45%). Critically, the HATU conditions failed to generate any cyclic tetrapeptides (**11f**, **11h**, **11g**) when the linear peptides lacked turn-inducing proline or N-methyl amino acid residues, yielding only oligomers. In contrast, β-thiolactone chemistry successfully produced the desired cyclopeptide scaffolds even at a higher concentration of 1 mM (compared to 0.5 mM for HATU coupling), achieving yields of 40–54%. The initial comparison experiments indicated the superior ability of the β-thiolactone-mediated generation of tetracyclic peptide scaffolds.

The scope and limitations of β-thiolactone-mediated tetracyclic peptides were further investigated. β-Thiolactone-mediated head-to-tail cyclization is a versatile strategy. The general synthetic approach to produce tetrapeptides bearing thiolactones is depicted in Fig. 4. Four different β-thiolactones derived from the amino acids Ala/Cys (**7a**), Val

(**7b**), Leu (**7c**), and Phe (**7d**) are individually coupled to tripeptide **12**, which features diverse amino acid residues[49], resulting in a variety of linear tetrapeptides **13**. Most thiolactone tetrapeptides are stable under acidic and neutral storage conditions for several weeks; an exception is cysteine-derived β-thiolactone peptides, which should be cyclized within three days. Notably, the thiol moiety from cyclized product **14** could be removed via an in situ postcyclization desulfurization step, leading to native cyclo-tetrapeptides **15**. L-Cyclo(Pro-Tyr-Pro-Val(SH)) **14a** was prepared in 68% yield with the addition of borax, and subsequent one-pot desulfurization furnished **1** in 57% yield over two steps. Cycles with a consensus sequence of L-cyclo(Pro-Xxx-Pro-Xxx) were also successfully prepared (Figs. 4, 2–4). In this case, tetrapeptides bearing either leucine- or phenylalanine-derived β-thiolactones were converted to L-cyclo(Pro-Leu-Pro-Leu(SH)) **2a** and L-cyclo(Pro-Phe-Pro-Phe(SH)) **3a** in comparable yields (56% and 58%, respectively). The thiol appendages could be swiftly removed under standard desulfurization conditions. Cyclic peptides **2** and **3** were obtained in 43% and 49% yields, respectively, and L-cyclo(Pro-Val-Pro-Val) **4** was generated in a lower yield (35%) due to the poor solubility of intermediate **4a**. Interestingly, placing a cysteine moiety between two prolines afforded cyclopeptide **14c** in good yield. Other amino acid side chains were investigated, and our methodology afforded wide functional group tolerance; residues such as Tyr, Asn, Trp, Thr, Met, and His did not interfere with the cyclization and led to decent yields. On the other hand, amino acids, such as Asp and Arg, caused the isolation yields to erode (**14d** and **14h**). Ring closure with N-methyl amino acids produced macrocycles **14l–q**. Furthermore, sequences bearing one proline residue or N-Me amino acid could be cyclized to furnish a range of desired products in good to excellent yields

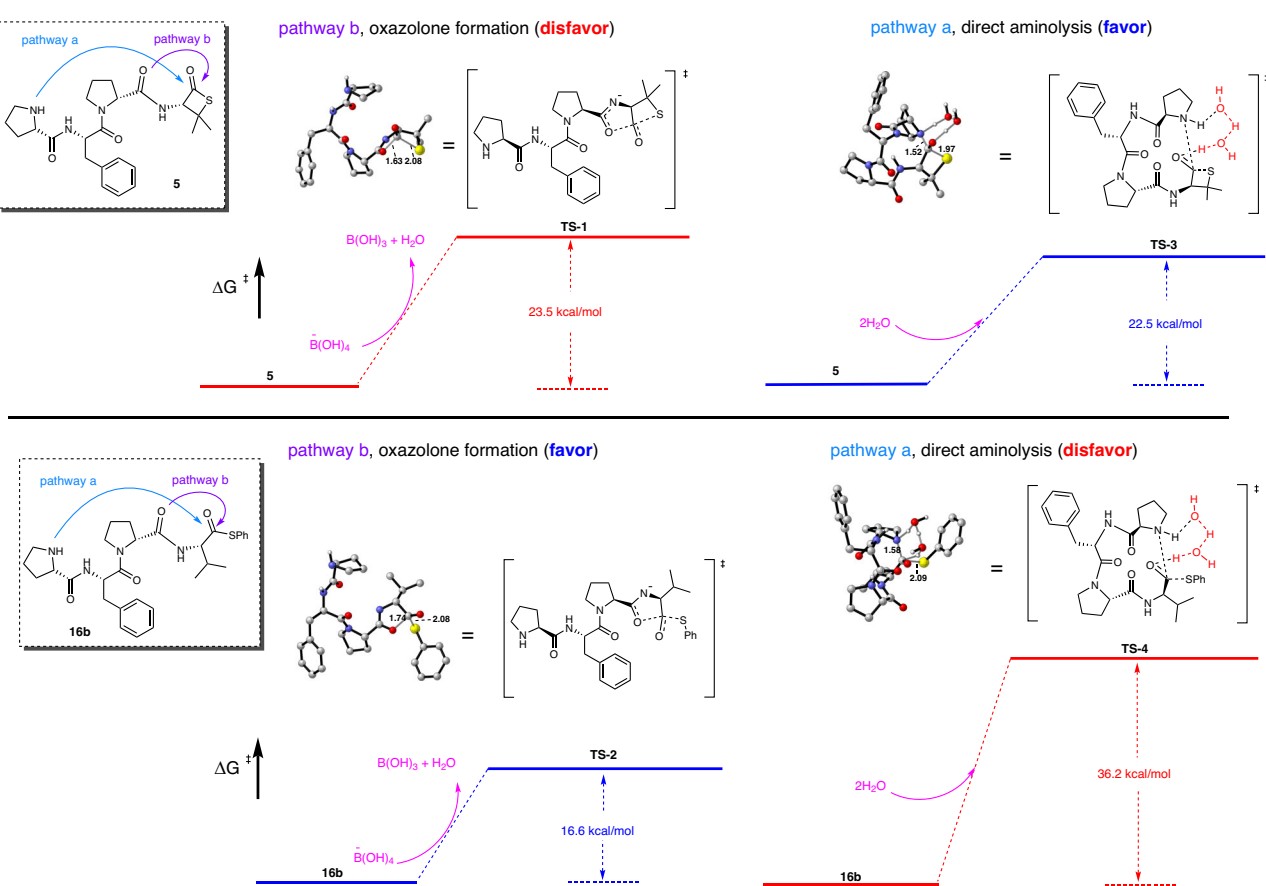

**Fig. 6 | Comparison of the Gibbs free energy barrier for different cyclization pathways.** Hydrogen atoms have been intentionally excluded for clarity in depicting the molecular structures.

(47–73%, **14n**–**r**). Notably, ring closure was achieved without known turn-inducing residues, such as Pro or N-methyl motifs, and **9f** and **14s**–**v** were prepared in mixed yields (21–54%). However, our methodology involves the following issues: initial attempts to produce **14i** were unsuccessful in the presence of unprotected lysine, as its side-chain ε-amino group outcompeted the proline amine and yielded a 13-membered ring exclusively. This problem was mitigated by introducing a Boc group to the lysine sidechain, and the desired substrate **14i** was obtained in good yield. Removal of the thiol and Boc groups afforded product **15i**. The cyclization of peptides with two or more adjacent prolines (Pro-Pro-Pro-Val/Ala-Pro-Pro-Val) failed to furnish the desired product, and only oligomer byproducts were observed. The cyclization of tetrapeptides bearing enantiomeric D-β-thiolactone at the C-terminus was unfruitful, and the dimeric octapeptides were isolated as the main products. The β-thiolactone-mediated approach is well suited for proteinogenic tetracyclic peptide synthesis. However, its effectiveness decreases significantly during the production of pentacyclic peptides, presumably due to limitations in borax-facilitated head-to-tail cyclizations. In conclusion, the β-thiolactone-mediated cyclization method involves a wide range of reaction scopes, accommodates various amino acid side chains, and enables the synthesis of diverse cyclic tetrapeptides. The protocol enabled efficient and reliable cyclization, as observed in the synthesis of cyclic tetrapeptides presented in Figs. 3 and 4. Intriguingly, we never observed epimerization during these investigations.

## C-terminus epimerization studies

Owing to their structural constraints, β-thiolactones exhibit heightened reactivity compared to conventional thioesters[38]. Consequently, the enhanced reactivity inhibits oligomerization and promotes

intramolecular cyclization, while also increasing the likelihood of oxazolone formation and epimerization. However, the absence of epimerization in the β-thiolactone-mediated cyclization suggested that the unique scaffold in β-thiolactones plays a crucial role in preventing epimerization. To investigate this topic, a comparative analysis was conducted between the β-thiolactone method and the direct aminolysis of regular thioesters, as depicted in Fig. 5a. Head-to-tail cyclization of precursor linear peptides **16a**, **16b**, and **16f** in aqueous buffer with borax yielded abysmal results, as a mere 3% yield was attained for **3**, and **11b** was completely epimerized at the C-terminus (15%). In the case of linear peptide **16f** without a turn inducer, cyclization to **11f** was not achieved, leading to complete hydrolysis and epimerization.

To further investigate the epimerization process in the absence of β-thiolactone, the cyclization of the linear peptide pair **16b** with L-Val-SPh and **d-16b** with D-Val-SPh thioesters was individually examined in aqueous buffer (Fig. 5b). Under borax conditions, cyclization of **16b** resulted in the formation of L-Val cyclopeptide **11b** and D-Val cyclopeptide **d-11b** as a mixture of isomers at a ratio of 1:3 (15% of **11b**). Notably, the ring closure of peptide **d-16b** (D-Val thioester) independently produced the same pair of isomers, **11b** and **d-11b**, in a 1:3 ratio. Both **16b** and **d-16b** underwent undesired intramolecular cyclization, generating identical azlactone/oxazolone **16** intermediates, which subsequently underwent enolization and eliminated the valine α-stereogenic center. The remaining reactive intermediate **16** underwent ring closure, yielding the cyclo-peptides **11b** and **d-11b** as a pair of isomers.

We propose that racemization is initiated by the formation of an azlactone/oxazolone during C-terminal activation. The torsion angle of the activated carbonyl ester (–SPh) results in rotational freedom,

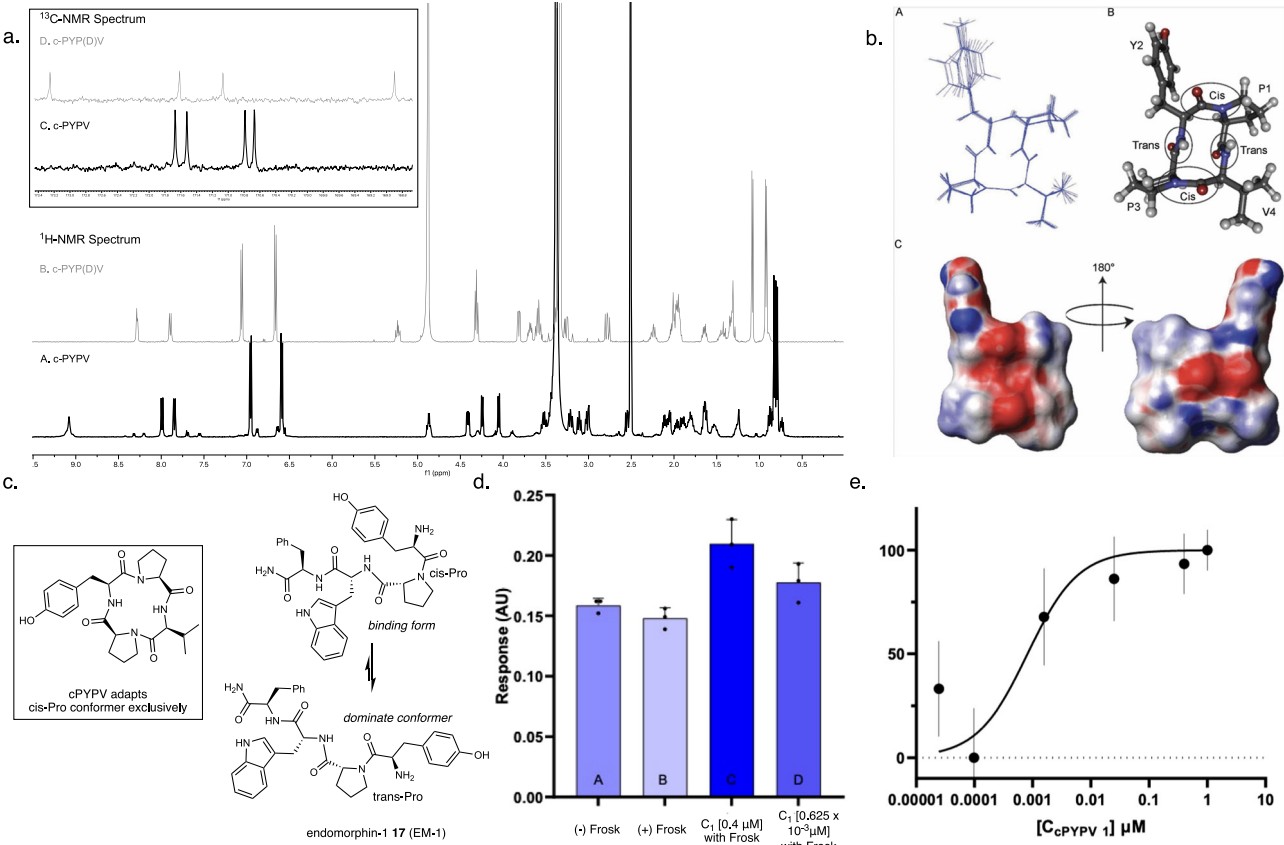

**Fig. 7 | Additional investigation of L-cyclo (PYPV).** Structure calculations were performed in CYANA version 3.98.5. (PDB code 6DNY). **a** Comparison of the ¹H-NMR spectra of Cyclo-(Pro-Tyr-Pro-Val) and Cyclo-(Pro-Tyr-Pro-(D)Val). A. ¹H-NMR spectrum of Cyclo-(Pro-Tyr-Pro-Val), B. ¹H-NMR spectrum of Cyclo-(Pro-Tyr-Pro-(D)Val), C. ¹³C-NMR spectrum of Cyclo-(Pro-Tyr-Pro-Val) from δ 168 to δ 174, D. ¹³C-NMR spectrum of Cyclo-(Pro-Tyr-Pro-(D)Val) from δ 168 to δ 174. **b** A. NMR ensemble overlay of 10 conformers showing the cyclic nature of the tetrapeptide Cyclo-(Pro-Tyr-Pro-Val). B. Ball and stick model of cyclo-[(L)Pro-(L)Tyr-(L)Pro-(L)Val] 1 showing cis–trans isomerization, Pro1 and Pro3 are in a cis orientation

while Tyr2 and Val4 are trans. C. Electros-static surface map of 1 showing positive, 0.5 kT/e (blue), and negative, −0.5 kT/e (red), regions of the electrostatic potential, where k = Boltzmann constant, T = temperature, and e = electron charge.
**c** Structure of 1 and cis–trans isomers of endomorphin-1 (EM-1) 17. **d** Accumulation of cAMP in human neuroblastoma cells (SK-N-SH) (measured using a cAMP ELISA-based colorimetric assay (Cayman, kit 581001). **e** Dose–response of accumulated cAMP following treatment with 10 μM Forskolin and different concentrations of 1. Column heights reported on the graph represent mean values, and error bars represent the SD.

enabling the molecule to align properly with the adjacent amide carbonyl and facilitating the formation of the azlactone (Insert I, simplified Newman projection). In contrast, the geometry of the four-membered ring moiety in β-thiolactone may inhibit epimerization, as depicted in Fig. 5c, which is analogous to the ability of proline to resist C-terminal epimerization[50,51]. The presence of the thietane ring restricts the rotation of the ψ torsion angle, preventing optimal overlap between the thiolactone carbonyl antibonding orbital and adjacent amide lone pair (III, simplified Newman projection). Additionally, the nearly flat geometry of the β-thiolactone induces 1,3 allylic strains between the adjacent amide carbonyl (O-1) and thiolactone α-hydrogen, orienting the thiolactone ring away from the amide carbonyl nucleophile (inset, II). Consequently, the lone pair of carbonyl O-1 cannot approach the C-2 antibonding orbital at the ideal Bürgi–Dunitz angle[52] of 107°, effectively inhibiting undesired epimerization.

## Computational studies

The results of computational simulations support the preference for β-thiolactone-mediated epimerization-resistant direct aminolysis over conventional approaches. In the examples of cyclization of **5** (β-thiolactone) and **16b** (phenyl thioester), the transition state configurations correspond to the two competing pathways (Fig. 6, inset), delineated

as direct aminolysis (a) and the oxazolone pathway (b). The oxazolone pathway involves the formation of a transition state resembling a tetrahedron (TS-1 and TS-2). Nucleophilic reactivity is increased in an aqueous environment with a buffer because the NH group undergoes partial deprotonation. The deprotonation process involves the transfer of a proton from the NH group of the molecule to the tetrahydroxyborate, likely originating from borax decomposition after solvation in water[53]. DFT calculations supported that this deprotonation process is endothermic; therefore, the energy barrier of the oxazolone pathway included contributions from the deprotonation steps. Our findings indicate that the energy barrier of standard thioester peptide **16b** is lower (TS-2, 16.6 kcal/mol) in this mechanism than direct aminolysis (TS-4, 36.2 kcal/mol), suggests that thioester peptide **16b** strongly prefers to undergo oxazolone intermediate formation. In contrast, the literature states that direct aminolysis may necessitate the presence of water molecules[54]. Therefore, the inclusion of water molecules could substantially lower the energy barrier for the reaction. Our investigative outcomes support this trend. Specifically, Fig. 6 reveals the potential energy surfaces (PESs) of pathway a, which includes two additional water molecules. Notably, based on the energy profile and inclusion of two water molecules, direct ring closure tends to occur by the 5-aminolysis mechanism with the β-thiolactone substituted intermediate, as evidenced by an energy barrier of

22.5 kcal/mol (TS-3). This barrier is lower than the corresponding value for the oxazolone pathway intermediate (TS-1 23.5 kcal/mol). Therefore, our results suggest that β-thiolactone substitution in **5** tends to favor the aminolysis mechanism, while regular thioester-substituted **16b** intermediates tend to promote adjacent carbonyl nucleophilic attack. The observed preference between the N-terminal amine and carbonyl group towards the C-terminus aligns with the principles of hard and soft acid–base theory (HSAB). The phenyl thioester is a softer acid due to carbonyl resonance, while β-thiolactone is regarded as a harder acid because its constrained structure limits resonance[55]. Conversely, the resonance of the amide at the Cterminal-1 position softens the amide carbonyl base[56], with the N-terminal primary amine serving as a harder base. As a result, the harder base of the N-terminal amine preferentially reacts with the harder acid of β-thiolactone, while the softer base of the amide carbonyl selectively combines with the softer phenyl thioester as an acid.

Overall, utilizing the epimerization-resisting direct aminolysis method for peptide ring formation results in notable distinctions and offers significant advantages over conventional cyclization approaches. This approach can address several problems, including the imbalanced formation of desired products and the occurrence of unwanted byproducts due to excessive C-terminal activation. β-Thiolactone facilitates efficient intramolecular aminolysis reactions while effectively preventing epimerization through conformational constraints. The reported method results in higher yields, shorter reaction times, fewer side products and more importantly, avoids epimerization. In contrast, the extant tactic generates products in low to no yields and involves significant oligomerization, while the direct aminolysis approach using thioesters results in complete epimerization (cf. Figs. 3 and 5).

### Spectroscopic analyses and biological assessments

Spectroscopically, cyclopeptides **2**–**4** exhibited symmetric geometries, as evidenced by their NMR spectra. The $^1H$ NMR spectra of cyclopeptides **2** and **4** displayed homotopic protons, indicating their symmetric nature. In the NMR spectrum of cyclopeptide **3**, accelerated interconversion between conformers was observed at an elevated temperature (65 °C). These symmetric NMR spectra provided strong evidence that the thiolactone-mediated cyclization did not induce racemization. Although cyclic tetrapeptides are structurally constrained[30,57], many macrocycles exhibit multiple conformers, especially for compounds that contain multiple prolines. Extensive NMR experiments, along with HPLC observation, confirmed that the cyclization did not produce epimerized isomers (see Supplementary Information VI for NMR experiment details). Since researchers have suggested that L-cyclo (Pro-Tyr-Pro-Val) **1** cannot be chemically prepared by the existing methods[36], subsequent extensive experiments were performed to confirm the identity of **1**. After numerous attempts and strategies, including derivatizing the sidechain of **1**, a single crystal of L-cyclo (Pro-Tyr-Pro-Val) was not obtained. Notably, Fig. 7a shows the distinct differences observed in the $^1H$ and $^{13}C$ NMR spectra of cPYPV and cPYP (D)V. The 1D $^1H$ and $^{13}C$ NMR spectra of peptide **1** revealed three distinct sets of major peaks, indicating the presence of three predominant conformers. The ratio of the three conformers was approximately 10:1:1, and the minor conformers were not identical to cPYP(D)V. These results further excluded the epimerization of L-β-thiolactone during the cyclization process. 1D and 2D homonuclear NMR analyses confirmed the major conformer structure of **1**, which was determined to be cyclo (cisPro-transTyr-cisPro-transVal), and all the amino acids were L-epimers (Fig. 7b).

To explore the application of synthetic cyclic peptides, we assessed their potential bioactivity, and multiple macrocycles, as shown in Fig. 4, were evaluated. Among them, the primary structure of L-cyclo-Pro-Tyr-Pro-Val **1** closely resembles that of the endogenous opioid tetrapeptide endomorphin-1 **17** (EM-1; Fig. 3c; sequence Tyr-Pro-Trp-Phe-NH$_2$). EM-1 is a highly selective agonist of the μ-opioid receptor (MOR) and could be used as an opioid drug substitute due to its analgesic effects[58]. Based on the SAR studies, the Tyr-Pro structure of **17** closely resembles opioid compounds, such as morphine, and a *cis*-proline configuration is necessary for μ-opioid receptor recognition[54]. The linear peptide EM-1 possesses less than 20% cis proline, and the cyclo-tetrapeptide proline adapts exclusively to the cis geometry. We expected that **1** should demonstrate superior MOR binding activity[59,60]. After exposure to **1** and the cAmp activator forskolin, the cAMP level in human neuroblastoma SK-N-SH changed, suggesting that the protease-resistant cyclic tetrapeptide is a potent MOR agonist with an EC$_{50}$ of 2.5 nM (Fig. 7d, e). Other cyclopeptides, **14a**, **9b** and **15e**, were evaluated along with **1**, and none of these molecules demonstrated superior activities (>55 μM for all), which underscores the importance of the cis-proline-tyrosine scaffold for effective GPCR binding.

In summary, this study presents a direct aminolysis tactic to chemically synthesize highly constrained proteinogenic tetracyclic peptides. The presence of a rigid four-membered ring is advantageous because it provides a significantly reactive C-terminus and conformationally locked geometry that resists oxazolone. The β-thiolactone-mediated direct aminolysis strategy simultaneously enables rapid head-to-tail amidation and effectively prevents epimerization and oligomerization. This approach challenges conventional notions that highly reactive C-terminal activators cannot robustly inhibit epimerization while facilitating the cyclization of tetrapeptides. Through the β-thiolactone method, proteinogenic tetracyclopeptides can be synthesized without restrictions on specific amino acid residues. The method utilizes a universal, head-to-tail direct coupling approach that tolerates a wide range of amino acid functional groups. In addition, this methodology was successfully used to synthesize several natural and synthetic constrained cyclopeptides, including the notable example of L-cyclo-(Pro-Tyr-Pro-Val), which was previously inaccessible. A computational approach provided support to preferably utilize the direct aminolysis pathway against oxazolone formation in thiolactone chemistry. Moreover, the study successfully demonstrated the biological activities of previously unattainable cyclopeptides. Specifically, the activity of L-cyclo-(Pro-Tyr-Pro-Val), which exhibits a potent agonistic effect (EC$_{50}$ = 2.5 nM) on the μ-opioid receptor, was established. This discovery underscores the pressing need for methods to facilitate interdisciplinary research on challenging synthetic biomolecules[61].

## Methods

### General procedures for peptide synthesis

**Preparation of β-thiolactone bearing linear tetrapeptides.** Sidechain fully protected tripeptide was prepared by solid phase peptide synthesis. To the mixture of tripeptide (1.0 equiv.), the β-thiolactone TFA salt (1.2 equiv.), and 3-hydroxy-1,2,3-benzotriazin-4-one (HOOBt) (1.2 equiv.) were added anhydrous CH$_2$Cl$_2$. The resulting solution was stirred at 0 °C for 2 min, and N-(3-Dimethylaminopropyl)-N-ethyl-carbodiimide (EDC) (1.2 equiv.) was added. The resulting mixture was stirred at 0 °C for another 1 h. The reaction was quenched with saturated NH$_4$Cl and extracted with CH$_2$Cl$_2$ (5 mL × 3), and the organic layer was dried over anhydrous Na$_2$SO$_4$. The Na$_2$SO$_4$ was filtered, and the solvent was removed via vacuum to afford a crude oily residue which was subjected to appropriate cocktail deprotection conditions at room temperature. After acid deprotection, the resulting solution was gently blown off by an argon stream to afford oily residue again. Finally, the oily residue was washed with cold diethyl ether to yield a white solid, which was dissolved in a mixture of acetonitrile and water and ready for HPLC purification after filtration. The desired β-thiolactone bearing tetrapeptides were purified through preparation HPLC and generated as a white powder after lyophilization.

**Cyclization of *C*-terminal β-thiolactone tetrapeptides.** 1.6 mM $Na_2HPO_4$ with $Na_2B_4O_7$ (1.0 equiv., 0.002 mmol) stock solution was prepared (pH = 7.4). To this stock solution, β-thiolactone bearing linear tetrapeptides (0.002 mmol) was added (final reaction concentration 1.0 mM), and the mixture was stirred at room temperature for 4 to 8 h under an argon atmosphere in the plastic reaction tube. The reaction was analyzed by LC–MS until the starting material was totally consumed. The solvent was removed by lyophilization, and the resulting white residue was dissolved in a mixture of acetonitrile and water prior to HPLC purification.

**One-pot desulfurization.** Upon completion of the cyclization as indicated by LC–MS analysis, to the reaction vessel was added 0.5 M Tris(2-carboxyethyl)phosphine hydrochloride (TCEP), ʟ-glutathione reduced (GSH) solution and radical initiator 2,2′-azobis[2-(2-imidazolin-2-yl) propane] dihydrochloride (VA-044) (0.1 M in degassed water). The reaction mixture was stirred at 37 °C until the completion with LC–MS monitoring, it was then quenched by adding $H_2O$/MeCN/AcOH (90:5:5, *v/v/v*) and purified by HPLC.

### Reporting summary

Further information on research design is available in the Nature Portfolio Reporting Summary linked to this article.

## Data availability

The solution NMR data generated in this study have been deposited in the protein database wwPDB, under data code ID: 6DNY. The supplementary information is available free of charge. Source data are present. All data are available from the corresponding author upon request. Source data are provided with this paper.

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

## Acknowledgements

Support for this work was provided by the National Institute of Health (R35 GM138336) to Q. Zhang and the NIH (2R01GM08500606A1) to A. Shekhtman. The National Natural Science Foundation of China (22122104 and 21933004) to X. Xue. We thank Profs. We thank David Crich (University of Georgia) and Ping Wang (Shanghai Jiao Tong University) for their helpful suggestions.

## Author contributions

Q.Z. conceived the project. A.S., X.X., and Q.Z. supervised the project. H.C., Y.W., X.F., N.S., Y.L., L.B., and Y.D. performed the experimental studies. Y.Z. performed the computational studies. A.S., X.X., and Q.Z. wrote the manuscript with support from all authors.

## Competing interests

The authors declare no competing interests.
