## [Peer Review File · Nature Communications]

Reviewers' comments:

Reviewer #1 (Remarks to the Author):

Ring closure of a linear to a cyclic tetrapeptides is often associated with low yields which is explained by the unfavorable relative orientation of amino and carboxy tertminus or by the ring strain of the product. The authors present a solution to the problem based on the nucleophilic opening of a thiolactone formed at the carboxy terminus of the peptide. This strategy of macrolactam formation requires the additional synthetic step of desulfurization, which is accepted because the tetrapeptides are otherwise inaccessible, as stated by the authors. A very interesting observation is the positive effect of borax in the macrocyclization reaction, which was systematically studied, although the exact role of the added salt remains speculative.

Cyclo-(Pro-Tyr-Pro-Val) is presented as a peptide that cannot be synthesized by existing lab methods. Lit 19 is cited to prove this claim (page 5, line 3). This review is not an original paper, moreover I cannot find this peptide in the review. In a recent paper by K. Fukase (Soc. Open Sci. 8: 201822. <https://doi.org/10.1098/rsos.201822>), I find the very similar cyclo-(Pro-Leu-Pro-Ile) and a handful of similar tetrapeptides cyclized with yields between 43 and 70% in a similar range with the methods used by the authors.

The manuscript is an interesting contribution to peptide chemistry and of interest to those skilled in the field. Borax-mediated amidation is an interesting observation, but as it stands the manuscript describes a technical advance that lacks deeper scientific significance.

I agree with the authors that the probability of epimerization during the cyclization reaction is low, but the methods used by the authors (NMR and modeling) are not suitable to prove this. Only chiral HPLC of the amino acids after complete hydrolysis of the peptide can rule out epimerization.

Minor problems:

The discussion of conformational isomerism of tertiary amide bonds is not acceptable. Please use the established physicochemical discussion of coalescence. Conformers do not "merge".

Page 5: All conformers merge into a single isomer at 60 °C ...; Supplement page 167: ...prove that the isomeric peaks are derived from structural conformational changes... and several more statements of the authors.

The spectrum at 50°C (Figure S16) has a very low signal-to-noise ratio. The minor isomer may be hidden in the noise without coalescence. This high temperature spectrum is not suitable to prove the presumed coalescence of two or more sets of signals of the title compound.

Cyclic tetrapeptides containing proline are difficult to draw, but they should at least be represented in the same way throughout the manuscript. The structures in which the amide carbon atoms point into the interior of the macrocyclic ring place the four oxygen atoms at an inappropriately close distance to each other.

Reviewer #2 (Remarks to the Author):

The authors describe a novel method for the synthesis of strained cyclic tetrapeptides using C-terminal thiolactone activation, followed by desulfurization. This method furnishes the cyclic tetrapeptides in moderate yields, without the epimerization that is a problem with other C-terminal activation methods. The authors demonstrate this method through the synthesis of a variety of cyclic tetrapeptides. The authors speculate as to why this method is resistant to epimerization. A cyclic analogue of endomorphin-1 is synthesised and tested for mu-opioid receptor activation.

The paper contains a number of formatting and typographical errors and is difficult to follow at various points. In addition to a thorough check of the spelling and for other errors (e.g. 'xx examples' in the abstract). The following points should be addressed if this paper is to be considered for publication:

The sequences for peptides 8 and 9 are not given in the manuscript, despite them being discussed in the text. Sequences for all numbered peptides should be clearly shown.

Desulfurization conditions are not shown despite this being discussed as part of the method. Conditions for all reactions need to be shown.

Figure 3A is labelled as a stereo view, when it is an overlay of the NMR structure ensemble.

Table 1 shows conditions for cyclization reactions with pH values given to one decimal place. The legend states that pH values were measured with pH paper. This is not sufficient to determine pH to

the stated level of accuracy. pH values should be stated to a whole pH unit or removed from the table.

The speculation about metal ions templating macrocycle formation is given without evidence. Given that there are number of other competing potential explanations this seems highly speculative and should be removed.

The borax mediated cyclization is stated to give almost exclusive formation of peptide 15, yet the yield is stated to be 65%. The wording here is inconsistent and should be amended to reflect the actual yield.

Table 2 states that peptides 1-4 are challenging natural products, but no reference is given for them being natural products. A reference should be given or the description changed.

The cyclizations shown in table are described as 'not substrate specific'. Given that there are considerable differences in yields, this is wrong and the wording should be changed.

The wording around the discussion of cyclization mechanism is wrong in places. Peptides 22 and 23 are described as being 100% epimerized- this would correspond to an inversion of configuration, which is clearly not the case. In the next line peptide 30 is described as being racemised, but only the C-terminal chiral centre is affected, so this cannot be considered a racemate. This wording should be fixed here and at any other points where chiral integrity is discussed.

The discussion of endomorphin 1 binding is very dubious. The 3D overlap between 1 and MOR bound form appears to refer to a bound cyclic mimic, rather than endomorphin 1 itself, though this is not clear from the text. The specific crystal structure which is being compared should be made clear, including the pdb code if available.

Overall this paper describes a promising method for producing cyclic peptides, but contains numerous errors. Some of the discussion is weak in terms of supporting evidence, especially for the cyclisation conditions and for the endomorphin 1 binding studies. This undermines the claims of the paper to substantial degree. Based on this I believe that this paper in its current form is unsuitable for publication in nature communications.

Reviewer #3 (Remarks to the Author):

The manuscript of Chen et al. describes the synthesis of cyclic tetrapeptides obtained by intramolecular aminolysis of linear precursors. The linear precursors feature a beta-thiolactone functionality at the C-terminus derived from beta-thiol amino acids. The thiol amino acid is desulfurized after the cyclization step. The authors showed that borax plays an important role in the cyclization process.

Cyclic tetrapeptides are extremely difficult to produce. Therefore, novel methods enabling to access such peptides without epimerization are highly needed.

The manuscript and supporting information are difficult to read by containing too many errors. Scheme 2 is also missing in the main manuscript.

The authors claim that cyclization proceeds without epimerization of the beta-thiolactone residue. The NMR of cyclic peptide 1 (cyclo(PYPV)) shows three sets of peaks. The authors claim that all conformers merge into a single isomer by heating at 60-65 °C. Since the peaks are not attributed, these data are difficult to interpret. Moreover, examination of the NMR data provided in the Supporting Information show a significant line broadening upon heating, and a significant reduction of the signal-to-noise ratio. The minor peaks are perhaps not seen just because of the reduction in S/N ratio.

It is very unclear how the authors can be so affirmative for the validity of the structure presented in Fig. 3 and the absence of epimerization. The chirality of the residue is an input of the method, and the algorithm minimizes the structure by taking into account the constraints obtained from NMR experiments. But the algorithm does not allow to change the chirality of the residues during minimization, which is fixed by the manipulator at the start.

Regarding the issue of epimerization, some important analyses and controls are missing.

For example, what is the D-amino acid content for each AA present in the cyclic peptides?

What happens if the beta-thiolactone is prepared with the D-AA? Does the peptide cyclize and is the final product the same or different from the cyclic peptide produced from L-beta-thiolactone?

To conclude, the authors do not provide the evidence for their claims regarding the lack of epimerization upon cyclization. The NMR part of the work must be clarified. The manuscript and Supporting Information must be corrected for missing scheme and mistakes.

Reviewers' comments:

Reviewer #1 (Remarks to the Author):

Ring closure of a linear to a cyclic tetrapeptides is often associated with low yields which is explained by the unfavorable relative orientation of amino and carboxy terminus or by the ring strain of the product. The authors present a solution to the problem based on the nucleophilic opening of a thiolactone formed at the carboxy terminus of the peptide. This strategy of macrolactam formation requires the additional synthetic step of desulfurization, which is accepted because the tetrapeptides are otherwise inaccessible, as stated by the authors. A very interesting observation is the positive effect of borax in the macrocyclization reaction, which was systematically studied, although the exact role of the added salt remains speculative.

Cyclo-(Pro-Tyr-Pro-Val) is presented as a peptide that cannot be synthesized by existing lab methods. Lit 19 is cited to prove this claim (page 5, line 3). This review is not an original paper, moreover I cannot find this peptide in the review. In a recent paper by K. Fukase (Soc. Open Sci. 8: 201822. <https://doi.org/10.1098/rsos.201822>), I find the very similar cyclo-(Pro-Leu-Pro-Ile) and a handful of similar tetrapeptides cyclized with yields between 43 and 70% in a similar range with the methods used by the authors.

Thanks for the insightful suggestion, the information regarding the challenging cyclo-tetrapeptide synthesis was cited from the following literatures.

1. The reference that directly claimed c-PYPV can not be synthesized: ref#18 in previous manuscript, #21 in this submission (*J. Pept. Res.* **1997**, 49, 67-73. doi: 10.1111/j.1399-3011.1997.tb01122.x). It described the following “We have not been able to prepare the tyrosinase inhibitor cyclo(Pro-Val-Pro-Tyr).”
2. The reference #17 in previous manuscript, # 20 in this submission. (*Org. Lett.* 2006, 8, 919–922. doi.org/10.1021/ol053095o) Claims “Despite the plethora of techniques to cyclize small peptides, a synthesis of cyclo-[(I)Pro-(I)Tyr-(I)Pro-(I)Val], a potent tyrosinase inhibitor, remains elusive because of the unfavorable transition state leading to the cyclic product”.

The reference provided by reviewer: K. Fukase (Soc. Open Sci. 8: 201822. <https://doi.org/10.1098/rsos.201822>) did indicated the successful synthesis of cPLPI under HATU/DIPEA/DMF/Dichloromethane conditions. Inspired by the report, we carried out a serial of comparison experiments which were illustrated in scheme 2. The Pro-Xxx-Pro-Xxx cyclization, facilitated by HATU, resulted in the successful formation of the corresponding cyclopeptide with yields comparable to those reported by K. Fukase. Thiolactone mediated cyclizations yielded analogous products with slightly improved yields compared to HATU chemistry. Significantly, the HATU approach resulted in much lower yields and considerable oligomerization in certain cases when cyclizing the Pro-Xxx-Xxx-Xxx sequences (scheme 2, **11d**, **11e**). Sequences lacking the turn-inducing proline residue were evaluated, and the HATU coupling method failed to produce cyclized peptides, resulting in the recovery of only oligomers (scheme 2, **11f-11h**), in contrast, the thiolactone chemistry generated desired cyclic peptides in decent yields (scheme 2, **9d-**

9h). Moreover, it is noteworthy that the utilization of a conventional phenyl thioester at the C-terminus during the preparation of cyclopeptides **11b** and **11f** led to complete epimerization (100%) at the C-termini (scheme 3a).

The experiments depicted in Schemes 2, 3a clearly demonstrated the distinct advantage of β -thiolactone chemistry, which effectively generates constrained tetracyclic peptides with relatively higher yields, absence of epimerization, and minimal oligomerization.

The manuscript is an interesting contribution to peptide chemistry and of interest to those skilled in the field. Borax-mediated amidation is an interesting observation, but as it stands the manuscript describes a technical advance that lacks deeper scientific significance.

We acknowledge that the previous submission required substantial improvement in terms of articulating the scientific significance of β -thiolactone chemistry. Consequently, we have extensively redrafted the paper to better highlight the importance of this manuscript, as outlined below:

This study employs a direct aminolysis approach, distinct from native chemical ligation (NCL), to form amide bonds between the amino group and C-terminus without the need for an N-terminal thiol group. Direct aminolysis offers wider applicability but may be susceptible to epimerization. In this study, experimental and computational evaluations revealed that without the presence of thiolactone thietane ring, conventional direct aminolysis leads to complete epimerization during the cyclization.

The main significance of this study is the identification of the dual role of the thietane ring motif. It effectively restricts rotation of the C-terminal carbonyl, preventing epimerization, while simultaneously maintaining high reactivity to produce constrained cyclic-tetrapeptides through the direct aminolysis tactic. This discovery represents a novel concept in peptide chemistry, as it has not been reported or suggested before.

I agree with the authors that the probability of epimerization during the cyclization reaction is low, but the methods used by the authors (NMR and modeling) are not suitable to prove this. Only chiral HPLC of the amino acids after complete hydrolysis of the peptide can rule out epimerization.

Minor isomeric peaks can be observed in the NMR spectra of certain reported sequences. However, the variant temperature experiments conducted are insufficient to definitively ascertain that these isomeric peaks do not originate from epimerization occurring at the C-terminus. By synthesizing isomeric cyclic sequences incorporating D-amino acids at the C-terminus, we successfully obtained NMR spectra that revealed noticeable discrepancies in profile between the cyclic tetrapeptides with L- and D-C-termini. These experimental validations strongly indicate that the observed isomeric peaks are not a result of epimerization induced by β -thiolactone chemistry. Furthermore, the variation in isomeric ratios of the same cyclopeptide observed upon altering the NMR solvent suggests that the solvent influences the isomeric ratios rather than the presence of

epimers. Matching NMR and LC-MS spectra from both the thiolactone route and the HATU coupling method were obtained for identical sequences (**3**, **11b-11e**), providing additional evidence that the observed isomeric peaks are not caused by C-terminus epimerization.

Minor problems:

The discussion of conformational isomerism of tertiary amide bonds is not acceptable. Please use the established physicochemical discussion of coalescence. Conformers do not "merge".

We have changed the narratives to "the accelerated interconversion of between conformers was observed at an elevated temperature (65 °C)." line 235.

Page 5: All conformers merge into a single isomer at 60 °C ...; Supplement page 167: ...prove that the isomeric peaks are derived from structural conformational changes... and several more statements of the authors.

The spectrum at 50°C (Figure S16) has a very low signal-to-noise ratio. The minor isomer may be hidden in the noise without coalescence. This high temperature spectrum is not suitable to prove the presumed coalescence of two or more sets of signals of the title compound.

We concur with the reviewer's concerns about the limitations of variant temperature NMR experiments. To overcome this limitation, we pursued an alternative method to determine the conformers post-cyclization. Interestingly, our findings indicate that all L-configurations of cyclic tetrapeptides yield conformers in the presence of prolines, corroborating similar observations in existing literature. (K. Fukase (Soc. Open Sci. 8: 201822. <https://doi.org/10.1098/rsos.201822>)). Our investigation revealed that the absence of proline or the incorporation of D-amino acids effectively eradicated the presence of NMR isomers. Furthermore, in the case of synthetic cyclic tetrapeptides containing a thiol (-SH) group, a single conformer was consistently observed, even in instances with multiple prolines. This phenomenon can be attributed to the bulkiness of the -SH group, which effectively locks the conformation of the cyclic structure (comparable to t-butyl group, see *Org. Lett.* **2013**, *15*, 2246–2249 doi.org/10.1021/ol400820y).

In our revised submission, we provide the NMR spectra of **S16** analogue, **14g** (prior to desulfurization), in Table 1. The spectra demonstrate a single set of peaks, indicating the absence of isomers. We also include other examples of cyclic tetrapeptides with a thiol (-SH) group in Table 1, which exhibit similar observations of a single set of peaks in their respective NMR spectra.

Cyclic tetrapeptides containing proline are difficult to draw, but they should at least be represented in the same way throughout the manuscript. The structures in which the amide carbon atoms point into the interior of the macrocyclic ring place the four oxygen atoms at an inappropriately close distance to each other.

We have redrawn the cyclopeptide structures entirely.

Reviewer #2 (Remarks to the Author):

The authors describe a novel method for the synthesis of strained cyclic tetrapeptides using C-terminal thiolactone activation, followed by desulfurization. This method furnishes the cyclic tetrapeptides in moderate yields, without the epimerization that is a problem with other C-terminal activation methods. The authors demonstrate this method through the synthesis of a variety of cyclic tetrapeptides. The authors speculate as to why this method is resistant to epimerization. A cyclic analogue of endomorphin-1 is synthesized and tested for mu-opioid receptor activation.

The paper contains a number of formatting and typographical errors and is difficult to follow at various points. In addition to a thorough check of the spelling and for other errors (e.g. 'xx examples' in the abstract). The following points should be addressed if this paper is to be considered for publication:

We sincerely apologize for the multiple errors in our original manuscript. The spell and erroneous issues have been fully addressed with completely redrafted manuscript.

The sequences for peptides 8 and 9 are not given in the manuscript, despite them being discussed in the text. Sequences for all numbered peptides should be clearly shown.

In the revised version, we have renumbered the sequences **8** and **9** as **S9** and **S10**, respectively. This information can be found in the supporting information (section IV.2). These sequences represent unsuccessful cyclization attempts, where an NCL-like approach initially resulted in the production of a polymer as the major product. Recognizing that this information was not directly relevant to the main narratives, we have deleted the reaction scheme 2. However, we inadvertently neglected to remove the description of the reaction scheme, which has now been completely deleted in the new draft.

Desulfurization conditions are not shown despite this being discussed as part of the method. Conditions for all reactions need to be shown.

Desulfurization conditions have been displayed in table 1, condition d.

Figure 3A is labelled as a stereo view, when it is an overlay of the NMR structure ensemble.

The label has been changed to NMR ensemble overlay. Please see update in figure 3b.

Table 1 shows conditions for cyclization reactions with pH values given to one decimal place. The legend states that pH values were measured with pH paper. This is not

sufficient to determine pH to the stated level of accuracy. pH values should be stated to a whole pH unit or removed from the table.

pH has been updated according to the pH Meter measurements.

The speculation about metal ions templating macrocycle formation is given without evidence. Given that there are number of other competing potential explanations this seems highly speculative and should be removed.

We have removed the speculation from the text.

The borax mediated cyclization is stated to give almost exclusive formation of peptide 15, yet the yield is stated to be 65%. The wording here is inconsistent and should be amended to reflect the actual yield.

We have changed the narratives and deleted the inconsistent statement.

Table 2 states that peptides 1-4 are challenging natural products, but no reference is given for them being natural products. A reference should be given or the description changed.

The natural products claims are not accurate and being removed from the table, and the description has been changed to “Challenging Cyclopeptide with Consensus Sequences“. Please see table 1 for detail.

The cyclization shown in table are described as 'not substrate specific'. Given that there are considerable differences in yields, this is wrong, and the wording should be changed.

We have deleted the words “not substrate specific”.

The wording around the discussion of cyclization mechanism is wrong in places. Peptides 22 and 23 are described as being 100% epimerized- this would correspond to an inversion of configuration, which is clearly not the case. In the next line peptide 30 is described as being racemised, but only the C-terminal chiral centre is affected, so this cannot be considered a racemate. This wording should be fixed here and at any other points where chiral integrity is discussed.

We have changed the discussion description to: “100% epimerized at C-terminus” in both scheme and text.

The discussion of endomorphin 1 binding is very dubious. The 3D overlap between 1 and MOR bound form appears to refer to a bound cyclic mimic, rather than endomorphin 1 itself, though this is not clear from the text. The specific crystal structure which is being compared should be made clear, including the pdb code if available.

The co-crystal structure of EM-1 and MOR is not available. The previous submission utilized computation simulated EM-1 binding scaffold (Ref #41 and 42) to overlay with cPYPV. As reviewer has pointed out, it is not very accurate illustration of EM-1. We subsequently deleted the Figure 4b and replaced it with a pair of EM-1 illustrating cis and trans proline isomerization in this submission (Figure 3c). Figure 3c highlights the significance of cis-proline in MOR agonist binding and illustrates the barrier that linear EM-1 encounters in adopting a cis-proline configuration.

Overall this paper describes a promising method for producing cyclic peptides, but contains numerous errors. Some of the discussion is weak in terms of supporting evidence, especially for the cyclisation conditions and for the endomorphin 1 binding studies. This undermines the claims of the paper to substantial degree. Based on this I believe that this paper in its current form is unsuitable for publication in nature communications.

Reviewer #3 (Remarks to the Author):

The manuscript of Chen et al. describes the synthesis of cyclic tetrapeptides obtained by intramolecular aminolysis of linear precursors. The linear precursors feature a beta-thiolactone functionality at the C-terminus derived from beta-thiol amino acids. The thiol amino acid is desulfurized after the cyclization step. The authors showed that borax plays an important role in the cyclization process.

Cyclic tetrapeptides are extremely difficult to produce. Therefore, novel methods enabling to access such peptides without epimerization are highly needed.

The manuscript and supporting information are difficult to read by containing too many errors. Scheme 2 is also missing in the main manuscript.

We apologize for the errors. We have extensively revised the manuscript, addressing all spelling errors and typos. Furthermore, the document has undergone thorough proofreading by a native speaker to ensure linguistic accuracy and clarity.

The missing Scheme 2 was a submission error, and it has been relocated to the supporting information in section IV.2. The sequences **8** and **9** have been appropriately renumbered to **S9** and **S10** in the supporting information as well. In the previous manuscript, Scheme 2 illustrates the unsuccessful cyclization observed when initially employing an NCL-like approach. The predominant product of this reaction was a polymeric polypeptide, which was not directly relevant to the main narratives and thus was removed from the new manuscript. We apologize for the oversight in deleting the corresponding description of Scheme 2, it has now been eliminated in the new draft.

The authors claim that cyclization proceeds without epimerization of the beta-thiolactone residue. The NMR of cyclic peptide 1 (cyclo(PYPV)) shows three sets of peaks. The

authors claim that all conformers merge into a single isomer by heating at 60-65 °C. Since the peaks are not attributed, these data are difficult to interpret. Moreover, examination of the NMR data provided in the Supporting Information show a significant line broadening upon heating, and a significant reduction of the signal-to-noise ratio. The minor peaks are perhaps not seen just because of the reduction in S/N ratio.

Thanks for the meaningful suggestion, the variant temperature NMR experiments were unable to provide conclusive evidence regarding the epimerization statement. So we have successfully synthesized cPYP(d)V using the HATU coupling strategy. By conducting ¹H and ¹³C NMR experiments, we have observed distinct spectroscopic differences between cPYPV and cPYP(d)V. Based on these results, we can confidently assert that L-cPYPV did not undergo C-terminus epimerization during the cyclization process.

It is very unclear how the authors can be so affirmative for the validity of the structure presented in Fig. 3 and the absence of epimerization. The chirality of the residue is an input of the method, and the algorithm minimizes the structure by taking into account the constraints obtained from NMR experiments. But the algorithm does not allow to change the chirality of the residues during minimization, which is fixed by the manipulator at the start.

We acknowledge the reviewer's criticism regarding the validation of the structure in Figure 3 solely based on solution NMR experiments. To address this concern, we synthesized cPYP(d)V and conducted a comparison of ¹H and ¹³C NMR spectra with the data obtained for cPYPV. The analysis revealed no spectroscopic overlap between the two isomers, providing further support for the proposed structure of cPYPV.

Regarding the issue of epimerization, some important analyses and controls are missing. For example, what is the D-amino acid content for each AA present in the cyclic peptides?

We appreciate you raising this important point about epimerization and D-amino acid content in the cyclic peptides. To address your concern, we conducted a thorough literature review, which revealed that the utilization of solid-phase peptide synthesis (SPPS) for introducing linear peptide precursors greatly diminishes the occurrence of residue epimerization. Additionally, the coupling between Pro-Xxx-Pro and β-thiolactone is not expected to cause racemization, as proline residues are known to be resistant to epimerization. We understood that the C-terminus is a potential site for epimerization. However, our NMR spectra analysis did not reveal the presence of D-amino acids in the cyclopeptide. To validate this, we compared the spectra with examples of cyclopeptides that do contain C-terminal D-amino acids and found no similar spectra profiles. Therefore, based on our analysis, there is no evidence to suggest the presence of D-amino acids in the cyclopeptide.

What happens if the beta-thiolactone is prepared with the D-AA? Does the peptide cyclize and is the final product the same or different from the cyclic peptide produced from L-beta-thiolactone?

Thanks for reviewer's constructive suggestion, we prepared D- β -thiolactone bearing tetrapeptide. Unfortunately, when D- β -thiolactone at C-terminus was subjected for cyclization, no ring closure products were observed. Instead, only dimeric compounds were produced. Please see main article, line 158, for detail.

To conclude, the authors do not provide the evidence for their claims regarding the lack of epimerization upon cyclization. The NMR part of the work must be clarified. The manuscript and Supporting Information must be corrected for missing scheme and mistakes.

REVIEWER COMMENTS

Reviewer #1 (Remarks to the Author):

The authors were not able to significantly improve the manuscript. I strongly recommend rejection of this paper.

The figures and schematics have been redrawn but are still confusing and unclear: 11b is shown twice in Scheme 3 in two different versions. Scheme 3 shows a total of 3 different versions of cyclic tetrapeptides. The numbering R, R1, R2, and R3 is used arbitrarily in Table 1, which is not even a table. R1 and R2 are always the same substituents (R1 = R2). Therefore, there is no need for different numbers. R and R3 are both part of the same pyrrolidine ring in the case of proline. The general side chains R are not indicated in any of the figures.

The orbitals of Scheme 3 are crude simplifications from which it is not possible to see either the "perfect overlap" or the "no overlap". It is not possible to see the orientation of the tiny orbital on the carbonyl in III. What type of orbital is shown here? The antibonding orbital of a carbonyl group (π^*) to which the nucleophile binds looks different. The explanations are oversimplified and inadequate to explain the complex stereoelectronic relationship.

The way it is argued here and the sloppiness with which tables and schemes are presented in the second version of the manuscript make me doubt that the macrolactamizations were also performed with the necessary experimental care.

Reviewer #2 (Remarks to the Author):

The authors have substantially addressed the issues highlighted in my earlier review.

The reworked manuscript does a much better job of describing the methods. Some ambiguity remains over the use of racemization/epimerization terminology, but this is improved and does not detract from the readability of the manuscript.

Minor errors:

Scheme 1 Caption is missing the 'd' label.

Scheme 2 labels 'dimmer' should read 'dimer'

Line 134 'Notable' should read 'Notably'

Line 201 'transitory' should read 'transition'

Line 206 DFT calculations are said to 'confirm' the endothermic nature of the deprotonation. It is perhaps better to say that they 'support', as the calculations are not definitive.

Reviewer #3 (Remarks to the Author):

The authors performed additional experiments to address the issues raised in my report and those of the other referees.

In particular, additional syntheses and experiments were done to clarify the extent of epimerization during thiolactone-mediated cyclization.

Given the importance of accessing this type of cyclic peptides, this manuscript might be suitable for publication in Nature Communications.

However, the revision must be significantly improved before publication.

Main points that must be addressed

1. A discussion regarding the reactivity of beta-thiolactones (see for example <https://doi.org/10.1021/jo9001728>) toward hard and soft nucleophiles in the manuscript is lacking.

2. Scheme 1. Indicating pH values of solvents in absence of the peptide is useless. What matters is the pH the reaction mixture. I can note also that the parameters are not varied independently of the others (pH vs buffer type or concentration, see also lines 88, 89...), hampering any useful conclusion about the impact of parameters.

Note d is not defined.

3. I'm surprised not to see more experiments aiming at understanding the role of borax in the cyclization process.

The authors report only yields. Besides the fact that no errors are provided for the yields, no kinetic experiments are provided that could shed light on the mechanism of the cyclization process. This is a weak point that must be addressed.

4. The manuscript is VERY difficult to read. The manuscript should be revised to improve the clarity of the text, figures and schemes.

5. The manuscript is still faced with significant wording and error issues, as well as a imprecise reporting of previous work & experimental conditions (not exhaustive list):

Line 19: C-terminus carboxylate should be C-terminus carbonyl

Line38 : doi:10.1016/j.tetlet.2008.05.049 & doi: 10.1021/ol053095o should be discussed and cited as attempts to overcome ring strain in tetrapeptide synthesis. 10.1021/ol053095o is highly relevant as an analogue of c(PYPV) has been produced using the copper catalyzed azide-alkyne cycloaddition reaction

Line 57: this thiolactone is not a cyclobutene ring

Line 77: Some of the solutions used as not buffers

Line 81, Scheme 1 and all documents: Please put a space between numbers and units.

Line 85: some of the items in the list are not metal salts

Line 106: β -thiolactone enables direct aminolysis is a superior approach compared to conventional coupling method.

Please revise

Other suggestions:

Did the authors tried X-ray racemic crystallography to solve the X-ray structure of the compounds?

REVIEWER COMMENTS

Reviewer #1 (Remarks to the Author):

The authors were not able to significantly improve the manuscript. I strongly recommend rejection of this paper.

The figures and schematics have been redrawn but are still confusing and unclear: 11b is shown twice in Scheme 3 in two different versions. Scheme 3 shows a total of 3 different versions of cyclic tetrapeptides. The numbering R, R₁, R₂, and R₃ is used arbitrarily in Table 1, which is not even a table. R₁ and R₂ are always the same substituents (R₁ = R₂). Therefore, there is no need for different numbers. R and R₃ are both part of the same pyrrolidine ring in the case of proline. The general side chains R are not indicated in any of the figures.

The misapplied substituent symbols R, R₁, R₂, R₃ were corrected in scheme 2, scheme 3 and table 1.

The orbitals of Scheme 3 are crude simplifications from which it is not possible to see either the "perfect overlap" or the "no overlap". It is not possible to see the orientation of the tiny orbital on the carbonyl in III. What type of orbital is shown here? The antibonding orbital of a carbonyl group (π^*) to which the nucleophile binds looks different. The explanations are oversimplified and inadequate to explain the complex stereoelectronic relationship.

The orbital diagrams were revised to demonstrate scenarios of "perfect overlap" and "no overlap," with two extra figures included to depict the orientation of carbonyls antibonding orbitals accurately. The π^* orbitals of thioester and thiolactone carbonyls were also adjusted to represent the correct geometry. Additionally, the term "simplified Newman projection" was added into the manuscript in lines 178 and 182 to emphasize that the explanation provided is a simplified representation.

The way it is argued here and the sloppiness with which tables and schemes are presented in the second version of the manuscript make me doubt that the macrolactamizations were also performed with the necessary experimental care.

In certain tables and schemes, there have been errors in using symbols for substituents (R-R₃). We believe these mistakes aren't reflective of the experiments' quality. The cyclization experiments were conducted meticulously, supported by comprehensive analysis via HPLC, LC-MS, ¹H and ¹³C NMR spectra for each analyzed compound.

Reviewer #2 (Remarks to the Author):

The authors have substantially addressed the issues highlighted in my earlier review.

The reworked manuscript does a much better job of describing the methods. Some ambiguity remains over the use of racemization/epimerization terminology, but this is improved and does not detract from the readability of the manuscript.

Minor errors:

Scheme 1 Caption is missing the 'd' label.

We have added the d label in the revised scheme 1 caption.

Scheme 2 labels 'dimmer' should read 'dimer'

We have corrected the errors in scheme 2.

Line 134 'Notable' should read 'Notably'

We have corrected "Notable" to "Notably" now in line 122.

Line 201 'transitory' should read 'transition'

We have changed the "transitory" to "transition" now in line 189.

Line 206 DFT calculations are said to 'confirm' the endothermic nature of the deprotonation. It is perhaps better to say that they 'support', as the calculations are not definitive.

We have changed the word "confirm" to "support" now in line 201.

Reviewer #3 (Remarks to the Author):

The authors performed additional experiments to address the issues raised in my report and those of the other referees.

In particular, additional syntheses and experiments were done to clarify the extent of epimerization during thiolactone-mediated cyclization.

Given the importance of accessing this type of cyclic peptides, this manuscript might be suitable for publication in Nature Communications.

However, the revision must be significantly improved before publication.

Main points that must be addressed

1. A discussion regarding the reactivity of beta-thiolactones (see for example <https://doi.org/10.1021/jo9001728>) toward hard and soft nucleophiles in the manuscript is lacking.

We appreciate reviewer's suggestion. The hard and soft acid-base theory (HSAB) could explain our observation well. A paragraph was added in line 209:

" The observed preference between the N-terminal amine and carbonyl group towards the C-terminus aligns with the principles of Hard and Soft Acid-Base Theory (HSAB). The phenyl thioester is a softer acid due to carbonyl resonance, while β -thiolactone is regarded as a harder acid because its constrained structure limits resonance. Conversely, the resonance of the amide at the Cterminal-1 position softens the amide carbonyl base, with the N-terminal primary amine serving as a harder base. As a result, the harder base of the N-terminal amine preferentially reacts with the harder acid of β -thiolactone, while the softer base of the amide carbonyl selectively combines with the softer phenyl thioester as an acid."

the following papers were cited for the HSAB theory:

Crich, D.; Sana, K. SN2-Type Nucleophilic Opening of β -Thiolactones (Thietan-2-ones) as a Source of Thioacids for Coupling Reactions *J. Org. Chem.* **2009**, *74*,3389-3393.

Kennitz, C.; Loewen, M. Amide Resonance" Correlates with a Breadth of C–N Rotation Barriers. *J. Am. Chem. Soc.* **2007**, *129*, 2521–2528.

2. Scheme 1. Indicating pH values of solvents in absence of the peptide is useless. What matters is the pH the reaction mixture. I can note also that the parameters are not varied independently of the others (pH vs buffer type or concentration, see also lines 88, 89...), hampering any useful conclusion about the impact of parameters.

Note d is not defined.

As pointed out by the reviewer, inaccurate pH measurements of solvents due to the absence of linear peptide. To address this, we now include pH measurements after peptide addition for each solution. Interestingly, most solution pH values remained largely unaffected. However, when linear peptide was dissolved in pure water solution in a Falcon tube, a significant decrease in pH occurred (from 6 to 3). This decline can be attributed to the acidic nature of linear peptide, as prep-HPLC produces purified linear peptide as a TFA salt. Conversely, in the case of thin borosilicate glass vials as reaction containers, the pH was minimally impacted following linear peptide addition. This observation may be attributed to the coating material on the glass wall, which likely neutralizes the TFA salt of the linear peptide. Which explains failed cyclization in falcon tube.

The formation of cyclic tetrapeptide was closely linked to the reaction pH. Optimal reaction outcomes were observed under slightly basic conditions (7.45). Additionally, we conducted cyclization experiments with varying concentrations of borax while maintaining the reaction pH, as well as pH variations while keeping 1 equivalent of borax constant. Our findings suggest that borax concentration does not significantly impact optimal reaction yields, and the catalytic amount of borax for reaction acceleration is difficult to detect. However, deviations from pH = 7.45, either above or below, resulted in inferior yields. A paragraph in line 90 has been incorporated to delineate our conclusion:

" In general, the pH of the reaction plays a crucial role in the success of cyclization. The presence of TFA additive in preparative HPLC eluents leads to the isolation of the linear peptide in its TFA salt form. The acidic nature of the linear peptide affects the reaction pH. The pH measured after adding the acidic peptide 5 to the Falcon tube is around 3, while it remains close to neutral in the thin borosilicate glass vial. The optimized conditions (1.0 equivalent of borax in PBS buffer, pH = 7.45 efficiently furnished cyclic tetrapeptide via direct aminolysis between proline and β -thiolactone, resulting in minimal polymer formation and the elimination of hydrolyzed products."

Note d has been defined in the caption of scheme 1

3. I'm surprised not to see more experiments aiming at understanding the role of borax in the cyclization process.

The authors report only yields. Besides the fact that no errors are provided for the yields, no kinetic experiments are provided that could shed light on the mechanism of the cyclization process. This is a weak point that must be addressed.

Kinetic studies involving ^{11}B NMR spectra of Borax (1 equivalent, 1 mM) in D_2O showed two peaks at 2

and 20 ppm, representing the boron complex and $\text{B}(\text{OH})_4^-$, respectively (Figure 1). Addition of an equal molar ratio of peptide did not induce any spectral shifts or new peaks during the 5-minute to 8-hour observation period. Overlaying the spectra before and after peptide addition revealed no discernible changes, indicating no interaction between the peptide and Borax.

The investigation into reaction rates progressed by altering the pH of the reaction. At $\text{pH}=7.45$, Borax presence enhanced cyclization yield by 10% (Figure 2a; 66% vs 56%) and reduced dimer byproduct formation to 2.1% from 4%. Shifting the pH higher or lower resulted in comparable yields but notably increased byproduct formation to 7.7% and 9.3%, respectively (Figure 2c, d).

The kinetic plot of the reaction, utilizing the equation $[-\ln(1 - \text{conversion})]$ (<https://www.mdpi.com/1996-1944/12/18/2995>) against time at various pH levels, exhibits distinct rate constants (Figure 3). Elevated pH levels correspond to higher rate constants, indicating an accelerated reaction rate.

Figure 2 pH Studies

Figure 3 Rate Disparity Under Different pH

Figure 4 Rate Studies Under Different Concentration

We investigated the influence of borax concentration on cyclization. Borax ($\text{Na}_2[\text{B}_4\text{O}_5(\text{OH})_4] \cdot 8\text{H}_2\text{O}$) supplies 4 equivalents of boron per molar molecule. Across different concentrations (0-1 equivalent of borax), overall product and byproduct yields showed no significant differences (Figure 4). However, the kinetic plot using varied concentrations revealed distinct rate trends (Figure 5). Utilizing GraphPad Prism

Figure 5 Rate Studies

6, linear regression and slope calculations were generated (Figure 6). Notably, at 1 molar equivalent of borax, cyclization was markedly accelerated with a slope of 0.00759, whereas 0.5 equivalent displayed a smaller slope of 0.002703. Conversely, there was no substantial rate disparity observed among 0.25, 0.125, and 0 equivalents of borax, with slopes around 0.0019. Overall, stoichiometric borax (1 equivalent) accelerated cyclization as a first-order reaction, albeit its catalytic effect was weak, and NMR spectra did not evidence borax-peptide binding. Borax improved overall yield by 10% and reduced byproduct formation by 50%.

Figure 6. Rate Slope Calculation

	0.5eq	0.25eq	0.125eq	0eq	1.0 eq
Best-fit values					
Slope	0.002703 ± 0.0008404	0.001965 ± 0.0004134	0.001998 ± 0.0003198	0.001887 ± 0.0003088	0.007591 ± 0.0006080
Y-intercept when X=0.0	-0.005406	-0.003931	-0.003996	-0.003773	-0.01518
X-intercept when Y=0.0	2.000	2.000	2.000	2.000	2.000
1/slope	370.0	508.8	500.5	530.0	131.7
95% Confidence Intervals					
Slope	0.0005421 to 0.004864	0.0009024 to 0.003028	0.001215 to 0.002780	0.001131 to 0.002642	0.006103 to 0.009079
Goodness of Fit					
Sy.x	0.5321	0.2618	0.2556	0.2468	0.4885
Is slope significantly non-zero?					
t	3.216	4.754	6.248	6.110	12.48
DF	5.000	5.000	6.000	6.000	6.000
P value	0.0236	0.0051	0.0008	0.0009	< 0.0001
Deviation from zero?	Significant	Significant	Significant	Significant	Significant
Data					
Number of X values	6	6	7	7	7
Maximum number of Y replicates	1	1	1	1	1
Total number of values	6	6	7	7	7
Number of missing values	2	2	1	1	1
Equation	Y = 0.002703*X - 0.005406	Y = 0.001965*X - 0.003931	Y = 0.001998*X - 0.003996	Y = 0.001887*X - 0.003773	Y = 0.007591*X - 0.01518

This information has been placed in supporting information in page 144.

4. The manuscript is VERY difficult to read. The manuscript should be revised to improve the clarity of the text, figures and schemes.

We apologize for the many shortcomings in the previous version of the manuscript. We have conducted a thorough revision, eliminated redundancies and streamlined the text, resulting in a reduction of over 300 words to enhance readability. Additionally, we have revised all figures, schemes, and tables to improve the clarity and coherence of the article. Finally, we resorted to the service from the Nature Editing Group for manuscript assistance, and it provided meaningful suggestions and edits.

5. The manuscript is still faced with significant wording and error issues, as well as a imprecise reporting of previous work & experimental conditions (not exhaustive list):

Line 19: C-terminus carboxylate should be C-terminus carbonyl

We have changed " C-terminus carboxylate" to " C-terminus carbonyl" throughout in the manuscript.

Line38 : doi:10.1016/j.tetlet.2008.05.049 & doi: 10.1021/ol053095o should be discussed and cited as attempts to overcome ring strain in tetrapeptide synthesis. 10.1021/ol053095o is highly relevant as an analogue of c(PYPV) has been produced using the copper catalyzed azide-alkyne cycloaddition reaction

We appreciate reviewer's valuable suggestions. The references were cited and described at line 64 as: "Synthetic attempts using conventional coupling strategies did not yield L-cyclo(Pro-Tyr-Pro-Val). Instead, cyclic tetrapeptide with D-amino acid residues or its triazole analogue was obtained. Furthermore, acyl migration as an alternative tactic failed to produce cyclic tetrapeptide due to structural constraints."

Line 57: this thiolactone is not a cyclobutene ring

We have changed "cyclobutene" to either "four-membered ring" or "thietane ring".

Line 77: Some of the solutions used as not buffers

We have changed the "buffers" to "solutions" where applicable.

Line 81, Scheme 1 and all documents: Please put a space between numbers and units.

We have added space between number and unit throughout the document.

Line 85: some of the items in the list are not metal salts

We have deleted word "metal".

Line 106: β -thiolactone enables direct aminolysis is a superior approach compared to conventional coupling method. Please revise

We have removed this inaccurate statement.

Other suggestions:

Did the authors tried X-ray racemic crystallography to solve the X-ray structure of the compounds?

The suggested experiment of growing racemic sample for single crystal was carried out but unsuccessful. The sequences cyclo-PFPV and d-cyclo-PFPV were both prepared and mixed. Multiple solvents were screened, unfortunately, none of these conditions provided quality single crystal for X-ray analysis. The NMR spectra of d-cyclo-PFPV has been included in supporting information.

REVIEWERS' COMMENTS

Reviewer #4 (Remarks to the Author):

[Note from the Editor: Reviewer #4 was asked to assess the response given to reviewer #3 who was not able to look over the manuscript again]

The authors have carried out experiments to address the issues raised by Reviewer #3 during the latest round of revision. I think most of the concerns have been adequately addressed except for a few minor issues. This manuscript's quality has been much improved so far and should be suitable for publication in Nature Communications, after perhaps one additional round of revision to correct the issues noted below:

1. The authors have done a good job measuring the pH values of the reaction mixtures per the original question #2 raised by Reviewer 3. However, the update claim/conclusion in Line 91 that "In general, the PH of the reaction plays a crucial role in the success of cyclization" is a bit inaccurate as clearly the yield was 62.0% (not low) when the pH is 7.14 at PBS with 1.0 equiv. Borax. A more appropriate claim could be that both the pH of the reaction and the presence of Borax play crucial roles.
2. I commend the authors' efforts on the extensive kinetics experiments in response to the previously raised question #3. Nevertheless, the authors should not only put these data in the Supporting Information, but also mention and discuss this in the main text, as the audience are always curious at reaction kinetics for any new reaction.
3. Following up with the previous question #3 on "errors for yields", the previously reported yields should carry error bars, and be represented as Mean or Average +/- SD or error.
4. Regarding the previously raised question #5, most typos and writings have been very well fixed except for Line 77. It looks like the word "buffers" was still there.

Reviewer #4 (Remarks to the Author):

[Note from the Editor: Reviewer #4 was asked to assess the response given to reviewer #3 who was not able to look over the manuscript again]

The authors have carried out experiments to address the issues raised by Reviewer #3 during the latest round of revision. I think most of the concerns have been adequately addressed except for a few minor issues. This manuscript's quality has been much improved so far and should be suitable for publication in Nature Communications, after perhaps one additional round of revision to correct the issues noted below:

1. The authors have done a good job measuring the pH values of the reaction mixtures per the original question #2 raised by Reviewer 3. However, the update claim/conclusion in Line 91 that "In general, the PH of the reaction plays a crucial role in the success of cyclization" is a bit inaccurate as clearly the yield was 62.0% (not low) when the pH is 7.14 at PBS with 1.0 equiv. Borax. A more appropriate claim could be that both the pH of the reaction and the presence of Borax play crucial roles.

We have updated the inaccurate statement in main text as the following:

" In general, the reaction yields are influenced by both pH and the addition of borax. The presence of TFA additive in preparative HPLC eluents leads to the isolation of the linear peptide in its TFA salt form. The acidic nature of the linear peptide affects the reaction pH. The pH measured after adding the acidic peptide **5** to the Falcon tube is around 3, while it remains close to neutral in the thin borosilicate glass vial. A slightly basic conditions enables the cyclization. Meanwhile, borax is critical for enhancing reaction yields, without the addition of borax, cyclization suffers reducing yield and substantial oligomer formation."

2. I commend the authors' efforts on the extensive kinetics experiments in response to the previously raised question #3. Nevertheless, the authors should not only put these data in the Supporting Information, but also mention and discuss this in the main text, as the audience are always curious at reaction kinetics for any new reaction.

We have placed two images in figure 2 and discuss in the main text regarding the kinetics experiments:

" To further under the role of borax reagent, preliminary ^{11}B NMR studies were conducted (Figure 2a). Borax ($\text{Na}_2[\text{B}_4\text{O}_5(\text{OH})_4]\cdot 8\text{H}_2\text{O}$) supplies 4 equivalents of boron per molar molecule. Borax (1 equivalent, 1 mM) in D_2O showed two peaks at 2 and 20 ppm, representing the boron complex and $\text{B}(\text{OH})_4^-$, respectively. Addition of an equal molar ratio of peptide **5** did not induce any spectral shifts or new peaks during the 5-minute to 8-hour observation period. Superimposing the spectra before and after peptide addition revealed no discernible changes, indicating no interaction between the peptide and Borax. The influence of borax concentration on cyclization was investigated. At different concentrations (0-1 equivalent of borax), product and byproduct yields showed no significant differences. Moreover, the kinetic plot of the reaction, utilizing the equation $[-\ln(1 - \% \text{yield})]$ against time at different equivalents of borax. the kinetic plot using

varied concentrations revealed distinct rate trends (Figure 2b). Slope calculations reveal that at 1 molar equivalent of borax, cyclization was markedly accelerated, whereas 0.5 equivalent borax displayed a flatter slope. Conversely, there was no substantial rate disparity observed among 0.25, 0.125, and 0 equivalents of borax. "

3. Following up with the previous question #3 on "errors for yields", the previously reported yields should carry error bars, and be represented as Mean or Average +/- SD or error.

The error bars have been added to all the plots according to reviewer's suggestions.

4. Regarding the previously raised question #5, most typos and writings have been very well fixed except for Line 77. It looks like the word "buffers" was still there.

We have deleted the word "buffers"